

# Targeting B4GALT7 suppresses the proliferation, migration and invasion of hepatocellular carcinoma through the Cdc2/CyclinB1 and miR-338-3p/MMP2 pathway

Chang Liu[1,2,*], Yuqi Jia[2,3,*], Xinan Zhao[2,3], Zifeng Wang[2,3], Xiaoxia Zhu[2,3], Chan Zhang[2,3], Xiaoning Li[2,3], Xuhua Zhao[2,3], Tao Gong[2,3], Hong Zhao[2,3], Dong Zhang[2,3], Yuhu Niu[2,3], Xiushan Dong[4], Gaopeng Li[4], Feng Li[5], Hongwei Zhang[6], Li Zhang[7], Jun Xu[7] and Baofeng Yu[2,3]

[1] Department of Biochemistry and Molecular Biology, Changzhi Medical College, Changzhi, China

[2] Department of Biochemistry and Molecular Biology, Shanxi Medical University, Taiyuan, Shanxi, China

[3] Key Laboratory of Cellular Physiology (Shanxi Medical University), Ministry of Education, China, Taiyuan, China

[4] Department of General Surgery, Shanxi Bethune Hospital, The Third Hospital of Shanxi Medical University, Taiyuan, China

[5] Central Laboratory, Shanxi Cancer Hospital; Shanxi Hospital Affiliated to Cancer Hospital, Chinese Academy of Medical Sciences; Cancer Hospital Affiliated to Shanxi Medical University, Taiyuan, China

[6] Department of Hematology, Shanxi Cancer Hospital; Shanxi Hospital Affiliated to Cancer Hospital, Chinese Academy of Medical Sciences; Cancer Hospital Affiliated to Shanxi Medical University, Taiyuan, China

[7] Department of General Surgery, The First Hospital of Shanxi Medical University, Taiyuan, China

[*] These authors contributed equally to this work.

Corresponding authors
Jun Xu, junxuty@163.com
Baofeng Yu,
shanxiyangchengdg@163.com

## ABSTRACT

**Background.** As a three-dimensional network involving glycosaminoglycans (GAGs), proteoglycans (PGs) and other glycoproteins, the role of extracellular matrix (ECM) in tumorigenesis is well revealed. Abnormal glycosylation in liver cancer is correlated with tumorigenesis and chemoresistance. However, the role of galactosyltransferase in HCC (hepatocellular carcinoma) is largely unknown.

**Methods.** Here, the oncogenic functions of B4GALT7 (beta-1,4-galactosyltransferase 7) were identified in HCC by a panel of *in vitro* experiments, including MTT (3-(4,5-dimethylthiazol-2-yl)-2,5-diphenyltetrazolium bromide), colony formation, transwell and flow cytometry assay. The expression of B4GALT7 in HCC cell lines and tissues were examined by qPCR (real-time quantitative polymerase chain reaction) and western blot assay. The binding between B4GALT7 and miR-338-3p was examined by dual-luciferase reporter assay.

**Results.** B4GALT7 encodes galactosyltransferase I and it is highly expressed in HCC cells and human HCC tissues compared with para-tumor specimens. MiR-338-3p was identified to bind the 3′ UTR (untranslated region) of B4GALT7. Highly expressed miR-338-3p suppressed HCC cell invasive abilities and rescued the tumor-promoting effect of B4GALT7 in HCC. ShRNA (short hairpin RNA) mediated B4GALT7 suppression reduced HCC cell invasive abilities, and inhibited the expression of MMP-2 and Erk signaling.

**Conclusion**. These findings identified B4GALT7 as a potential prognostic biomarker and therapeutic target for HCC.

# INTRODUCTION

Hepatocellular carcinoma (HCC) is the second most common cause of cancer deaths worldwide (*Llovet et al., 2022*; *Shi et al., 2021a*; *Shi et al., 2021b*; *Sas et al., 2022*). The liver tumor micro-environment (TME) is more complex than other types of cancer, as HCC mainly develops because of chronic inflammation and fibrotic tissue background (*Feng et al., 2022*). The TME consists of tumor cells, stromal cells and proteins within the extracellular matrix (ECM) (*Feng et al., 2022*). The ECM compositions can change based on the needs of tumor microenvironment (*Kang et al., 2022*). Considering the low survival rate, and high rates of recurrence and metastasis for HCC, identifying the molecular function of ECM proteins is urgently needed for understanding tumorigenesis.

A total of 90% of HCC patients develop in the context of liver damage and deterioration of liver function leads to elevated proteoglycan (*Dituri et al., 2022*; *Lujambio & Maina, 2021*). Proteoglycans (PGs) are molecules that consist of the protein core and glycosaminoglycan (GAG) chains. PGs are also a significant component of the ECM and regulate cell–cell and cell–matrix interactions (*Dituri et al., 2022*). GAG chains are attached to the serine residue of the protein core *via* a tetrasaccharide linkage region (GlcA $\beta$-1,3-Gal- $\beta$1,3-Gal- $\beta$1,4-Xyl- $\beta$1,3-) (*Mihalic Mosher et al., 2019*). B4GALT7 (beta-1,4-galactosyltransferase 7) encodes galactosyltransferase I (or UDP-galactose: O-xylosylprotein $\beta$1,4-D-galactosyltransferase) that is involved in the attachment of two galactose residues to xylose in the biosynthesis of the linkage region (*Arunrut et al., 2016*; *Mihalic Mosher et al., 2019*; *Salter et al., 2016*; *Sandler-Wilson et al., 2019*). Therefore, mutations in B4GALT7 lead to deficient production of proteoglycans (*Guo et al., 2013*; *Mihalic Mosher et al., 2019*; *Salter et al., 2016*; *Sandler-Wilson et al., 2019*). Mutations in B4GALT7 cause skeletal dysplasia, Ehlers-Danlos syndrome and Larson of Reunion Island syndrome (LRS), since B4GALT7 is correlated with the initiation of glycosaminoglycan side chain synthesis of PGs (*Mihalic Mosher et al., 2019*; *Arunrut et al., 2016*; *Sandler-Wilson et al., 2019*; *Delbaere et al., 2020*; *Caraffi et al., 2019*). The differential expression and prognostic value of B4GALT7 have been observed in glioblastoma and myeloma cells (*Zhang et al., 2021a*; *Bret et al., 2009*). However, the specific regulatory mechanism for B4GALT7 is largely unknown. Meanwhile, two members of the B4GALT gene family, B4GALT1 (beta-1,4-galactosyltransferase 1) and B4GALT5 (beta-1,4-galactosyltransferase 5), have been reported to be involved in the development of MDR (multidrug resistance) of human leukemia cells by regulating the Hh (hedgehog) signaling and the expression of P-gp (p-glycoprotein) and MRP1 (MDR-associated protein 1) (*Zhou et al., 2013*). B4GALT4 has been reported to promote microtubule spindle assembly in HCC by inducing the expression of PLK1 and RHAMM (*Dai, Wang & Gao, 2022*).

Here we investigated the expression of B4GALT7 in HCC cells and tissue, and its possible correlation with prognosis of HCC patients. The cell functions and the relative biochemical pathways upon B4GALT7 suppression will be also investigated to provide a comprehensive picture of its role in HCC cell behavior.

## MATERIAL AND METHODS

### Cell lines and tumor tissues

Human HCC cell lines Huh-7, HepG2, SMMC-7721 and SK-Hep-1 were purchased from Cell Bank of Chinese Academy of Sciences (Shanghai, China). Human hepatocyte cell line HL-7702 and human HCC cell line SNU-423 were purchased from American Type Culture Collection (ATCC, USA). They were maintained in DMEM (Huh-7, HepG2 and SK-Hep-1) or RPMI-1640 (SMMC-7721, SNU-423 and HL-7702) medium with 10% fetal bovine serum. All cell lines used in the study were tested and authenticated using short tandem repeat (STR) matching analysis. 10 pairs of HCC tissues and corresponding para-tumor specimens were collected from the Affiliated Tumor Hospital of Shanxi Medical University (Shanxi, China). Written informed consents were obtained from all participants before surgery. Collections and use of tissue samples were approved by the ethics committee of the Affiliated Tumor Hospital of Shanxi Medical University (approval number: KY2023017) and were in accordance with the Declaration of Helsinki.

### Transfection

The short hairpin RNA (shRNA)-B4GALT7 and an empty vector were designed by GenePharma (Shanghai, China). The lentiviral vector plasmid used was LV3 (H1/GFP&Puro). Puromycin (4 μg/ml) was applied to select stable cell lines using shRNA vector to mediate B4GALT7 suppression. The three shRNA sequences targeting B4GALT7 were as follows:

shRNA-B4GALT7-1: 5′-GCAACAGCACGGACTACATTG- 3′;
shRNA-B4GALT7-2: 5′-GCCTGAACACTGTGAAGTACC- 3′;
shRNA-B4GALT7-3: 5′-GCACTGTCCTCAACATCATGT- 3′.
LV3 NC: 5′-TTCTCCGAACGTGTCACGT- 3′ (NC: negative control).

The miR-338-3p inhibitor and mimics were purchased from GenePharma, and were transfected into SNU-423 and SK-Hep-1 cells using siRNA-mate (GenePharma, Shanghai, China). The sequence information is shown in Table 1. Plasmid DNA (pEX-3/B4GALT7) with the restriction enzyme cutting site XhoI/EcoRI was obtained from GenePharma. Transfection was conducted according to the manufacturer's instructions.

### Cell proliferation, colony formation, migration, and invasion assays

MTT (Solarbio, Beijing, China) was conducted to assay cell proliferation. Absorbance at 492 nm was examined on consecutive four days using a BioTek microplate reader (Winooski, VT, USA). For the colony formation assay, the colonies were stained with 0.5% crystal violet and photographed. Absorbance at 595 nm ($OD_{595}$) was determined with a BioTek microplate reader. Each experiment was carried out three times. For the wound healing assay, a scratch was made to the monolayer formed by indicated cells in 6-well

**Table 1  Sequence information.**

| Name | Sense sequence (5′–3′) | Antisense sequence (5′–3′) |
|---|---|---|
| miR-338-3p mimics | UCCAGCAUCAGUGAUUUUGUUG | ACAAAAUCACUGAUGCUGGAUU |
| NC mimics | UUCUCCGAACGUGUCACGUTT | ACGUGACACGUUCGGAGAATT |
| miR-338-3p inhibitor | CAACAAAAUCACUGAUGCUGGA | |
| Inhibitor NC | CAGUACUUUUGUGUAGUACAA | |

**Notes.**

NC, negative control.

plates. Cells were further maintained without FBS for 48 h. The wound was photographed using light microscope and the wound healing area was calculated by Image J software. For the migration and invasion assay, the upper chambers were coated with or without 100 μL of Matrigel (1:8 mixed with FBS-free medium; Corning, New York, USA). 5–8 × 10⁴ indicated HCC cells were seeded in the upper chamber of transwell plates (8 μm pore size; Corning) without serum. Medium with 10% FBS was filled into the lower chamber. Cells on the bottom chamber were fixed, stained, and counted in five randomly selected fields using light microscope after 48 h.

## Dual-luciferase reporter assay

The GP-miRGLO-B4GALT7 WT (wild-type) plasmids and its corresponding mutant-type (mut) plasmids were designed by GenePharma. The above luciferase vectors, miR-338-3p mimics or miR-338-3p NC was co-transfected into HEK-293T or SNU-423 cells using Lipofectamine 2000 (Invitrogen). The dual luciferase reporter gene assay kit (GenePharma) was conducted to detect the renilla and firefly luciferase activities after incubation for 48 h.

## RNA extraction and real-time PCR

TRIzol reagent (Takara, Beijing, China) or miRNA Isolation Kit (Omega Bio-Tek, Guangzhou, China) was performed to extract total RNA. PrimeScript™ RT reagent Kit with gDNA Eraser (Takara) or Mir-X miRNA First-Strand Synthesis Kit (Takara) was performed to reversely transcribe cDNA from mRNA and miRNA. qRT-PCR was performed to calculate the mRNA levels by the $2^{-\Delta\Delta Ct}$ method using TB Green® Premix Ex Taq™ II (Takara). mRNA and miRNA expression levels were normalized to *β-actin* and small nucleolar RNA *U6*, respectively. The qRT-PCR primer sequences are shown in Table 2.

## Flow cytometry analysis of cell apoptosis and cell cycle

For analysis of cell apoptosis, the indicated HCC cells (1 × 10⁶ cells/mL) were incubated with 10 μL 7-AAD, 500 μL binding buffer and 5 μL Annexin V-APC for 15 min at 37 °C in the dark. For analysis of cell cycle, the indicated HCC cells were harvested, fixed in 70% ethanol, and incubated with RNase A and propidium iodide (PI) for 1 h at room temperature. The apoptosis rate and cell cycle were examined with an Agilent NovoCyte flow cytometer (Agilent, Santa Clara, USA). Each experiment was carried out three times.

| Table 2 | Primers and sequences. |
|---|---|
| Primers | Sequences (5′–3′) |
| B4GALT7-F | GGGAATCACAACTGGGTACAAGA |
| B4GALT7-R | CACATGGTACTTCACAGTGTTCAGG |
| $\beta$-Actin-F | TGGCACCCAGCACAATGAA |
| $\beta$-Actin-R | CTAAGTCATAGTCCGCCTAGAAGCA |
| hsa-miR-338-3p-F | CAGCATCAGTGATTTTGTTGAAA |
| hsa-miR-338-3p-R | Universal Reverse Primer (Takara) |
| U6-F | GGAACGATACAGAGAAGATTAGC |
| U6-R | TGGAACGCTTCACGAATTTGCG |

## Western blot analysis

Total proteins were extracted from the indicated HCC cells in RIPA buffer (Beyotime, Shanghai, China) and quantified using the BCA protein quantitation kit (Boster Biotechnology, Wuhan, China). Proteins in 60 µg samples were separated by 10% SDS-PAGE and transferred onto polyvinylidene fluoride (PVDF) membranes (Millipore, Burlington, MA, USA). The primary antibodies included B4GALT7 (1:500; NBP1-88652) from Novus Biologicals (Shanghai, China), MMP-2 (1:1,000; ab92536) from abcam (Cambridge, UK), and p44/42 MAPK (Erk1/2) (137F5) (1:1,000; #4695), Phospho-p44/42 MAPK (Erk1/2) (Thr202/Tyr204) (D13.14.4E) (1:1,000; #4370), Akt (pan) (C67E7) (1:1,000; #4691), Phospho-Akt (Ser473) (D9E) (1:1,000; #4060), E-Cadherin (24E10) (1:1,000; #3195), N-Cadherin (D4R1H) (1:1,000; #13116), Vimentin (D21H3) (1:1,000; #5741), Phospho-Chk2 (Thr68) (1:1,000; #2661), Phospho-Wee1 (Ser642) (D47G5) (1:1,000; #4910), Phospho-cdc2 (Tyr15) (10A11) (1:1,000; #4539), Cyclin B1 (D5C10) (1:1,000; #12231), phosphor-ATM (Ser1981) (D6H9) (1:1,000; #5883), Phospho-Histone H2A.X (Ser139) (20E3) (1:1,000; #9718) from Cell Signaling Technology (Danvers, MA), and $\beta$-actin (1:2,500; TA-09) from ZSGB-Biotechnology (Beijing, China). The membranes were visualized using an enhanced chemiluminescent (ECL) blot detection system (Transgene, Beijing, China) after the primary antibodies were incubated by anti-mouse or anti-rabbit secondary antibodies.

## Statistical analysis

Statistical analyses were performed using Student's $t$-test or one-way ANOVA by SPSS 19.0 statistical software. $P < 0.05$ was set as statistically significant.

## RESULTS

### Clinical relevance of B4GALT7 expression in HCC cancer patients

B4GALT7 was upregulated in HCC tissues compared to para-tumor specimens in three GEO datasets (GSE14520, GSE25097, GSE84402) (Fig. 1A, $P < 0.001$) and the TCGA database using the UALCAN portal ($P < 0.001$, Fig. 1B) (*Chandrashekar et al., 2017*). Consistently, upregulated B4GALT7 expression ($n = 90$) correlated with shorter survival probability in HCC patients ($P = 0.0032$, Fig. 1C). We further validated B4GALT7 expression levels in 10 paired human HCC tissues by western blotting (Fig. 1D) and

qPCR (Fig. 1E). B4GALT7 was overexpressed in seven (70%) HCC tissues compared with paired para-tumor specimens (Figs. 1D and 1E). B4GALT7 was mainly located in the cytoplasm and HCC tissues demonstrated stronger B4GALT7 staining than the paired para-tumor specimens (Fig. 1F), as revealed in the Human Protein Atlas database (https://www.proteinatlas.org/). We further applied the TIMER2.0 database (http://timer.comp-genomics.org/) to identify the expression landscape of B4GALT7. B4GALT7 was highly expressed in a large number of cancer tissues compared to para-tumor specimens (Fig. 1G).

## B4GALT7 suppression reduces HCC cell proliferation *in vitro*

We then examined the endogenous expression levels of B4GALT7 in five HCC cell lines by qPCR (Fig. 2A) and western blotting (Fig. 2B). B4GALT7 was highly expressed in SNU-423, SMMC-7721, SK-Hep-1, HepG2 and Huh-7 cells compared with normal liver cell HL-7702 (Figs. 2A and 2B). SNU-423 and SK-Hep-1, with the highest B4GALT7 expression levels, were chosen for further investigation. To examine the molecular mechanism by which B4GALT7 is associated with HCC, the SNU-423 and SK-Hep-1 cells were transfected with shRNA vectors to mediate B4GALT7 suppression. The green fluorescence intensity in both SNU-423 and SK-Hep-1 cells was above 80% (Fig. 2C). B4GALT7 was significantly downregulated in the above two cell lines by qPCR (Fig. 2D) and western blotting (Fig. 2E). ShRNA mediated B4GALT7 suppression in SNU-423 and SK-Hep-1 cells reduced cell proliferation rates (Figs. 2F–2G). However, no significant cell apoptosis was observed (Fig. 2H). Collectively, down-regulation of B4GALT7 reduces HCC cell proliferative abilities, but does not promote significant apoptosis *in vitro*.

## Down-regulation of B4GALT7 arrests the cell cycle at the G2/M phase

Then, we examined whether DNA was damaged after shRNA mediated B4GALT7 suppression by measuring the expressions of DNA damage markers, including ataxia-telangiectasia mutated (ATM) and H2AX (*Li et al., 2021b*; *Sharma & Almasan, 2021*). The phosphorylation of ATM and H2A.X was increased after shRNA mediated B4GALT7 suppression (Fig. 3A) and was reduced after plasmid pEX-3/B4LGAT7 mediated B4GALT7 overexpression (Fig. 3B). B4GALT7 expression was further rescued using plasmid pEX-3/B4GALT7 in SNU-423 transfected with shB4GALT7 (Fig. 3B). More cells stayed in the G2 phase for both cell lines after shRNA mediated B4GALT7 suppression, suggesting that B4GALT7 regulated the progression from G2 to M phase (Fig. 3C). Chk2 phosphorylation at Thr68 was significantly elevated after shRNA mediated B4GALT7 suppression (Fig. 3D). ShRNA mediated B4GALT7 suppression markedly promoted phosphorylation of Cdc2 at Tyr15 and induced the levels of cyclin B1 (Fig. 3D), which was assumed to extend the time for cells to fix DNA damages. ShRNA mediated B4GALT7 suppression in SNU-423 and SK-Hep-1 cells markedly promoted the phosphorylation of Wee1 at Ser642 (Fig. 3D). B4GALT7 overexpression rescued the cell cycle arrest caused by B4GALT7 suppression (Fig. 3E). Collectively, these results indicated that the ATM-Chk2-Cdc2/cyclin B1 pathway was involved in the G2/M cell cycle arrest caused by B4GALT7 suppression.

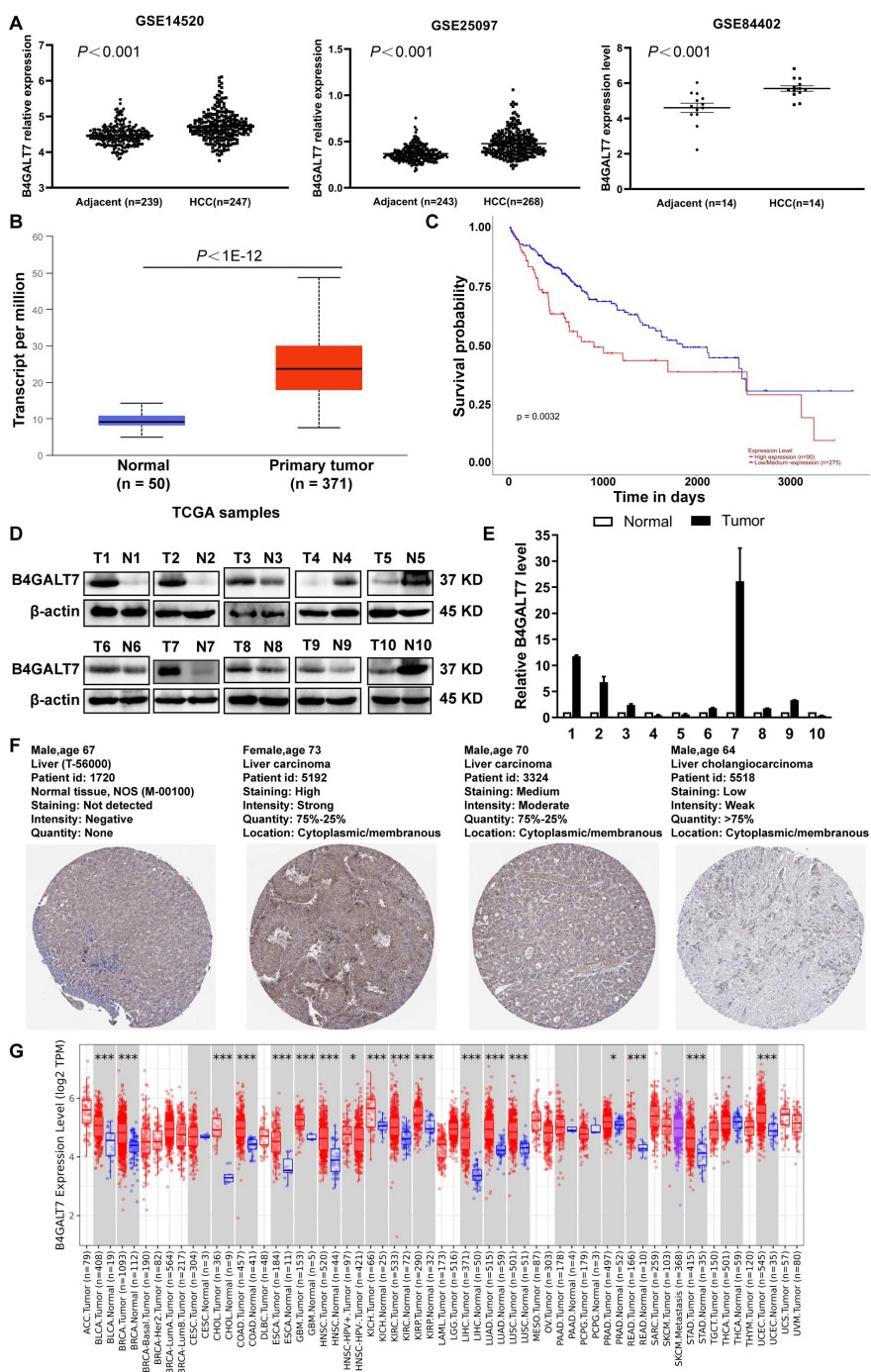

**Figure 1 B4GALT7 is highly expressed in HCC tissues.** (A) B4GALT7 expression levels in three GEO datasets, GSE14520, GSE25097 and GSE84402. (B) B4GALT7 levels in the TCGA database. (C) Survival probability of HCC patients with different expression of B4GALT7. The expression of B4GALT7 was analyzed by (D) western blotting and (E) real-time PCR in 10 pairs HCC samples and para-tumor specimens. (F) Representative immunohistochemical staining results of B4GALT7 based on the HPA database. (G) The expression landscape of B4GALT7 in the TIMER2.0 database.

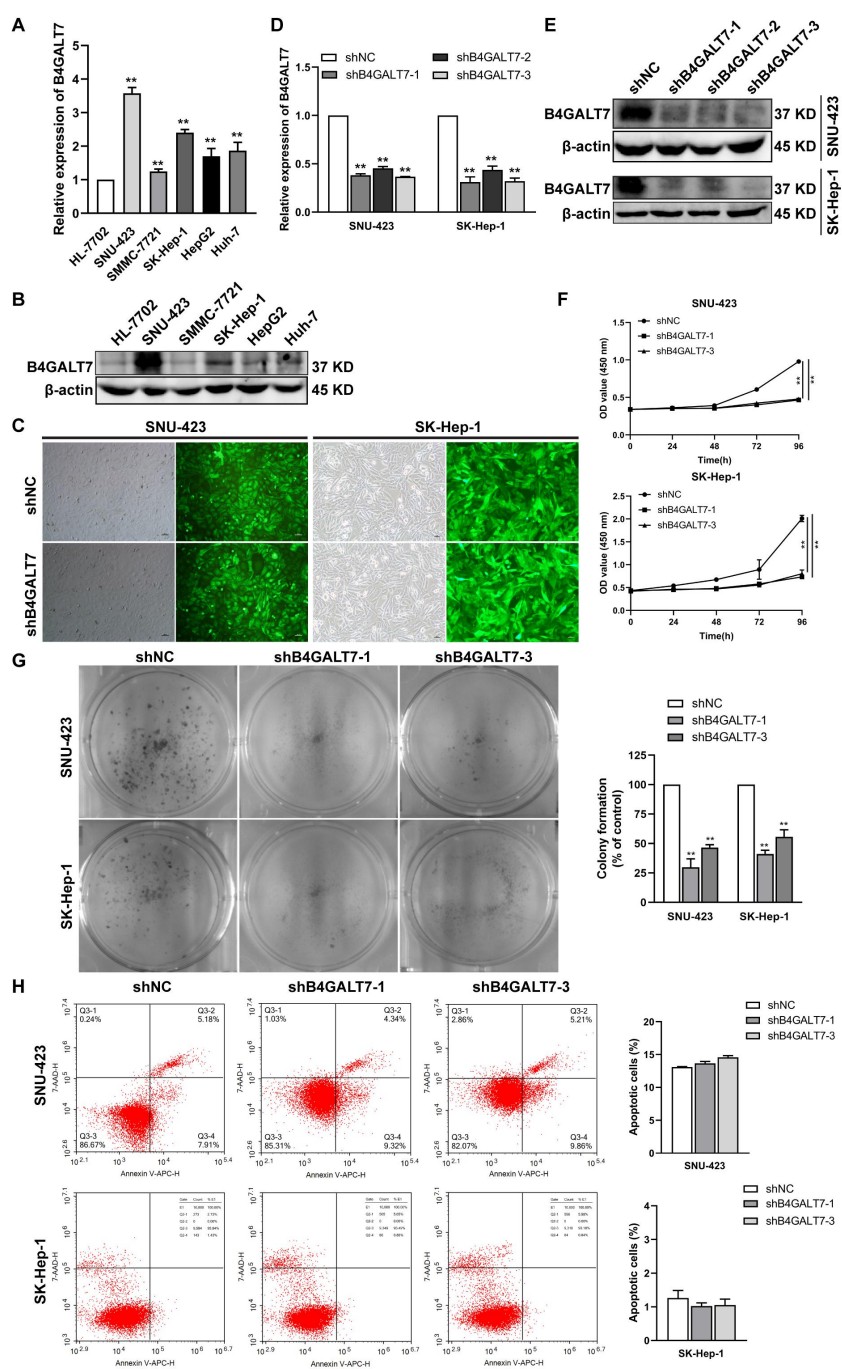

**Figure 2  Down-regulation of B4GALT7 inhibits HCC cell proliferative abilities *in vitro*.** (A) qPCR and
(B) Western blotting analysis of the expression of B4GALT7 in HCC cells (SNU-423, SMMC-7721, SK-
Hep-1, HepG2, Huh-7) and normal liver cell HL-7702. (C) Representative pictures of the green fluorescence intensity of HCC cells after transfected with shRNA vectors to mediate B4GALT7 inhibition. (D)
qPCR and (E) Western blotting analysis of B4GALT7 expression in HCC cells after transfected as in C.
Down-regulation of B4GALT7 inhibits the proliferative abilities of HCC cells (SNU-423, SK-Hep-1) determined by (F) MTT assay and (G) Colony formation assay. (H) Representative pictures of flow cytometry analysis of apoptosis stained with Annexin V-APC and 7-AAD in HCC cells transfected as in C. Scale
bar: 100 μm. Mean ± SD for three independent experiments are demonstrated. *, $P < 0.05$; **, $P < 0.01$.

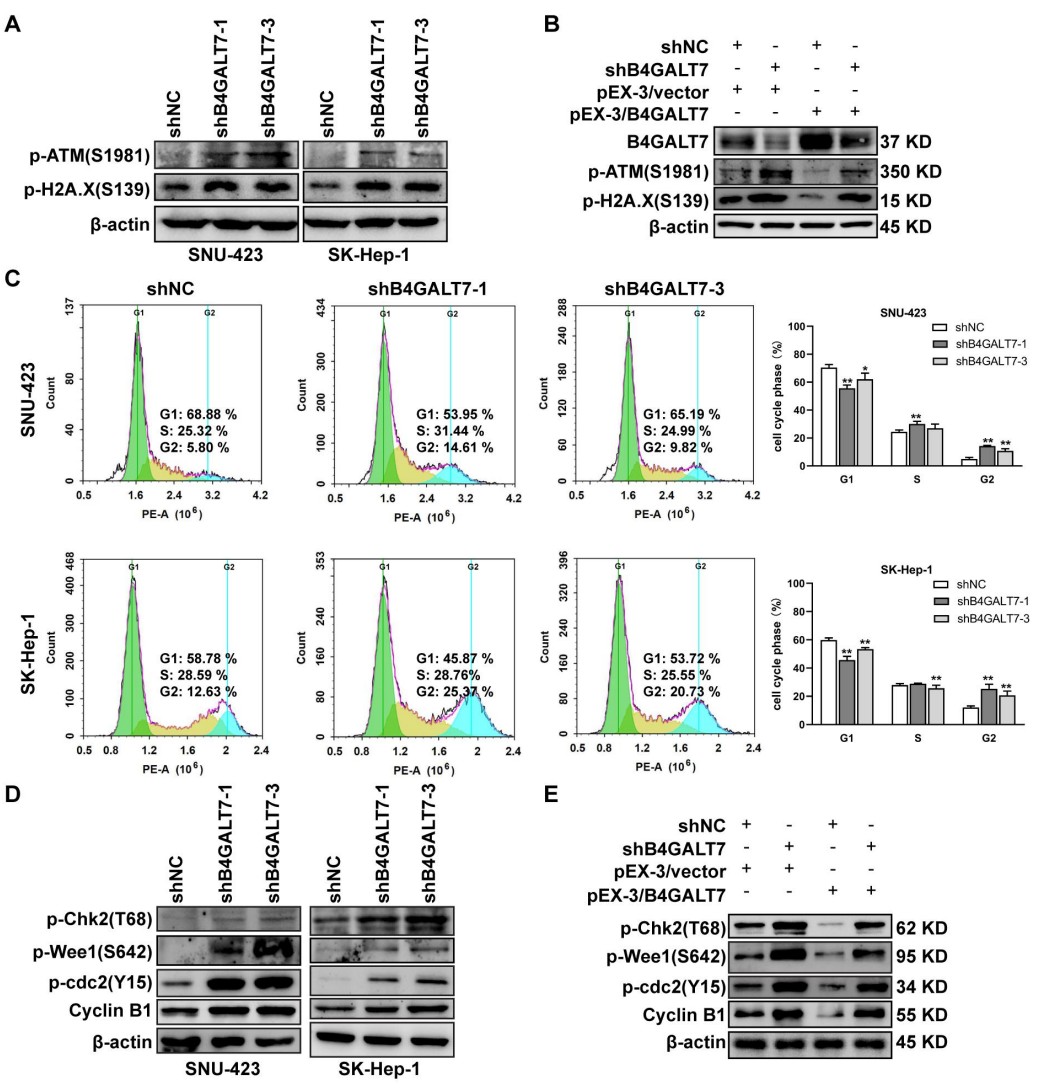

**Figure 3 Down-regulation of B4GALT7 results in DNA damage and arrests the cell cycle at the G2/M phase.** (A–B) p-ATM and p-H2A. X protein levels in SNU-423 and SK-Hep-1 cells transfected with shB4GALT7 or shNC, and further transfected with pEX-3/B4GALT7 or pEX-3/vector. (C) Cell cycle analysis in SNU-423 and SK-Hep-1 cells after shRNA mediated B4GALT7 inhibition. Mean ± SD for three independent experiments are demonstrated. (D–E) p-Chk2, p-Wee1, p-cdc2 and cyclin B1 protein levels in B4GALT7-downregulation SNU-423 and SK-Hep-1 cells, and further transfected with pEX-3/B4GALT7 or pEX-3/vector. *, $P < 0.05$; **, $P < 0.01$.

## B4GALT7 interacts with miR-338-3p in HCC cells

To examine the mechanism of B4GALT7 in modulating cell proliferative and invasive abilities, the online software TargetScan (http://www.targetscan.org/vert_72/) and miRPathDB v2.0 (https://mpd.bioinf.uni-sb.de/) were applied to screen for candidate miRNAs that might regulate B4GALT7. There is a potential 8mer binding site in the 3′ UTR of B4GALT7 for miR-338-3p (Fig. 4A). Low expression of miR-338-3p in SK-Hep-1 and SNU-423 cells was validated by qPCR analysis (Fig. 4B). MiR-338-3p suppressed

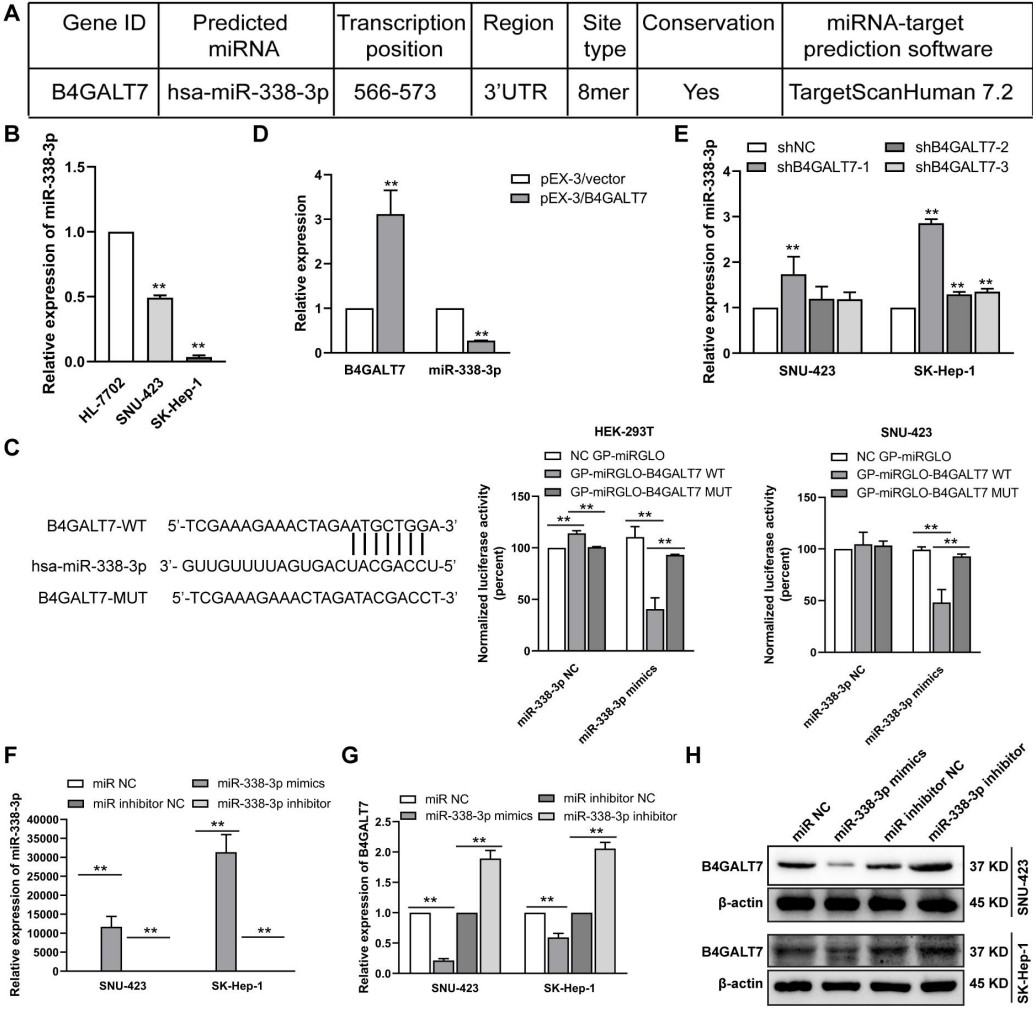

**Figure 4** **The reciprocal suppression effects of B4GALT7 and miR-338-3p.** (A) The potential interaction between B4GALT7 and miR-338-3p predicted by TargetScan. (B) Expression levels of miR-338-3p in HCC cells (SNU-423, SK-Hep-1) and normal liver cell HL-7702. (C) Dual luciferase reporter assay demonstrated the luciferase activities in HEK-293T and SNU-423 cells following the indicated transfection. (D–E) Expression levels of miR-338-3p in SNU-423 and SK-Hep-1 cells with B4GALT7 overexpression and after shRNA mediated B4GALT7 inhibition. (F–G) qPCR and (H) Western blotting analysis of B4GALT7 expression in SNU-423 and SK-Hep-1 cells after transfected with miR-338-3p mimics and inhibitor. *, $P < 0.05$; **, $P < 0.01$.

the luciferase activity of the B4GALT7-WT vector in the HEK-293T and SNU-423 cells, but not the B4GALT7-MUT vector, confirming that miR-338-3p targeted the 3′ UTR of B4GALT7 (Fig. 4C). B4GALT7 overexpression reduced miR-338-3p level (Fig. 4D) and shRNA mediated B4GALT7 suppression elevated miR-338-3p level in HCC cells (Fig. 4E). Overexpression of miR-338-3p (Fig. 4F) suppressed both mRNA (Fig. 4G) and protein expression levels (Fig. 4H) of B4GALT7, and miR-338-3p inhibition elevated both mRNA (Fig. 4G) and protein expression levels (Fig. 4H) of B4GALT7, implying that miR-338-3p degrades B4GALT7 mRNA by targeting its 3′ UTR.

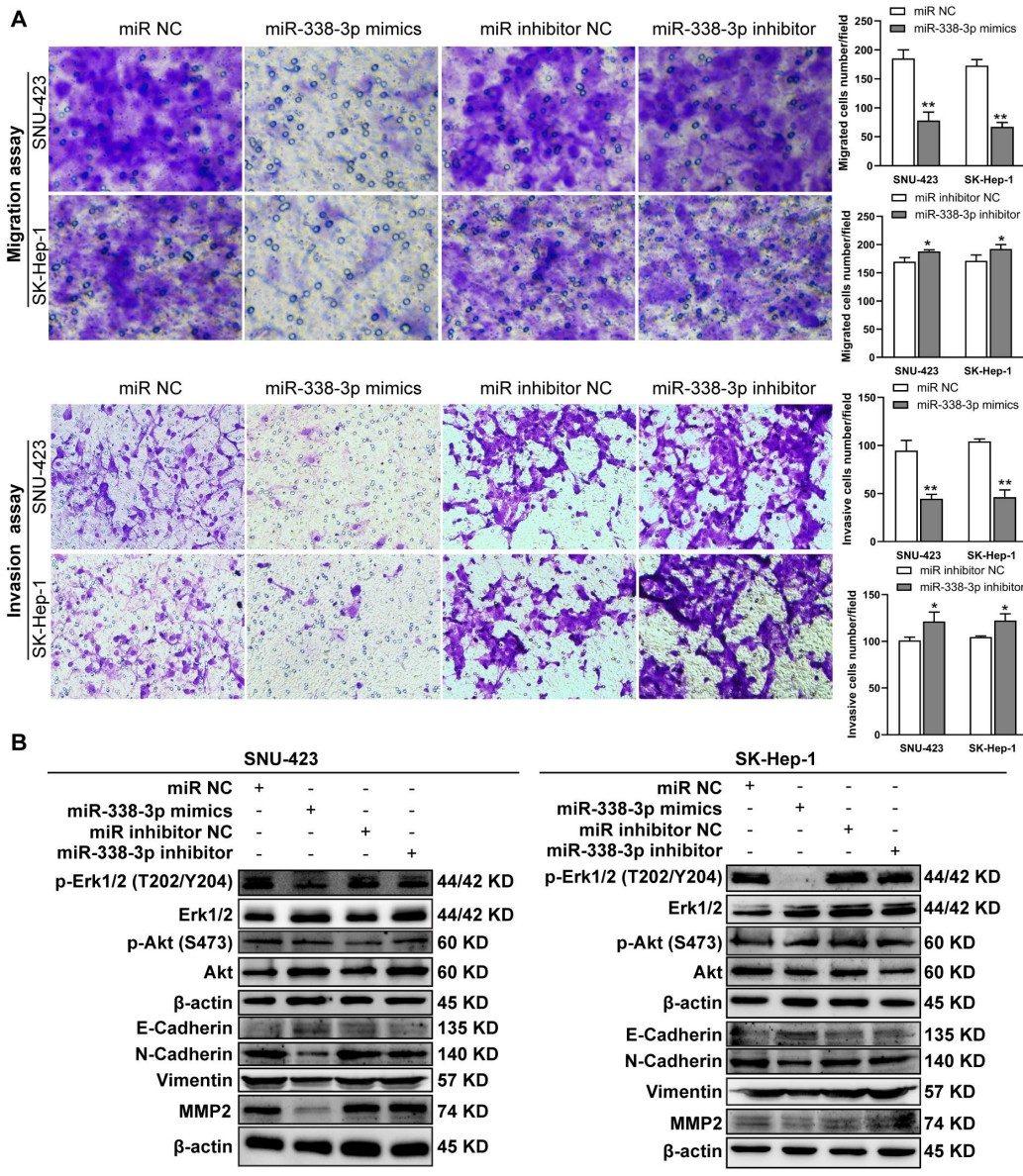

**Figure 5** **MiR-338-3p overexpression in HCC cells reduces cell migration and invasion.** (A) Matrigel-free and matrigel-based transwell assays revealed the effect of miR-338-3p on invasive abilities of SNU-423 and SK-Hep-1 cells. (B) Western blotting assay revealed the EMT marker protein expression and the phosphorylation status of signaling proteins in HCC cells transfected with miR-338-3p mimics and inhibitor. $\beta$-actin was used as the internal control. Scale bar: 100 μm. Data were shown as the mean ± SD. *, $P < 0.05$; **, $P < 0.01$.

Previous reports have shown that miR-338-3p is involved in the EMT (epithelial-mesenchymal transition) in HCC and other malignant tumors (*Li et al., 2021a*; *Lu et al., 2019*; *Song et al., 2020*; *Li et al., 2019*). We found that miR-338-3p mimics reduced HCC cell invasive abilities (Fig. 5A); and suppressed the phosphorylation of Erk (Fig. 5B), MMP2 and the expression of mesenchymal markers (N-cadherin and vimentin), whereas elevated

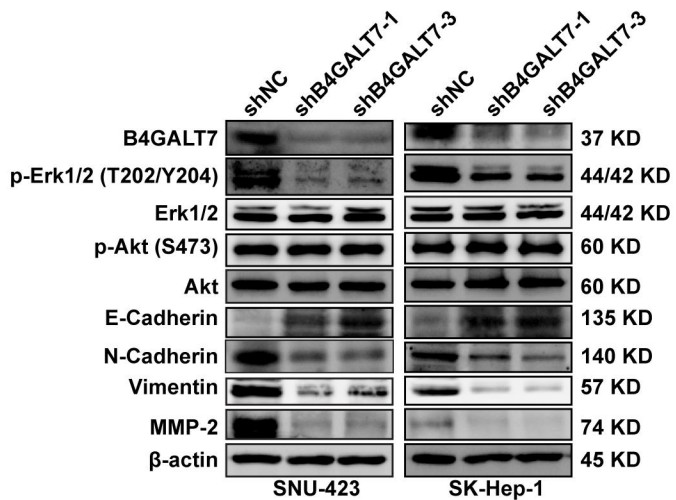

**Figure 6** Western blotting analysis of the phosphorylation status of signaling proteins and EMT marker proteins in B4GALT7-downregulation HCC cells. $\beta$-actin was used as the internal control for total proteins.

the expression of epithelial marker E-cadherin (Fig. 5B). However, the phosphorylation of Akt was not affected with miR-338-3p overexpression in the above two cell lines (Fig. 5B).

## B4GALT7 suppression reduces HCC cell migration and invasion *in vitro*

ShRNA mediated B4GALT7 suppression reduced the phosphorylation of Erk, MMP2 and the expression of mesenchymal markers (N-cadherin and vimentin), whereas elevated the expression of epithelial marker E-cadherin (Fig. 6). However, the phosphorylation of Akt was not affected after shRNA mediated B4GALT7 suppression in the above two cell lines (Fig. 6). Then, SK-Hep-1 and SNU-423 cells were transfected with different shRNAs/miR-338-3p inhibitors as demonstrated in Fig. 7A. We found that shRNA mediated B4GALT7 suppression suppressed the migrative and invasive abilities of HCC cells, whereas miR-338-3p inhibitor significantly rescued these phenotypes (Figs. 7A–7B). The expression levels of MMP2, N-cadherin, vimentin and E-cadherin were rescued after miR-338-3p inhibitor was co-transfected (Fig. 7C). In contrast, B4GALT7 overexpression induced HCC cell invasive abilities (Fig. 8A); and elevated the expression of MMP2 and the mesenchymal markers (N-cadherin and vimentin), whereas reduced epithelial marker E-cadherin (Fig. 8B). The invasion stimulative phenotypes were rescued after miR-338-3p mimics were co-transfected (Fig. 8A), and the expression levels of MMP2, N-cadherin, vimentin and E-cadherin were reversed (Fig. 8B). The expression levels of MMP2 and EMT marker proteins were reversed after transfection with plasmid pEX-3/B4GALT7 in SNU-423 with shRNA mediated B4GALT7 suppression (Fig. 8C). Collectively, these results suggested that miR-338-3p rescued the tumor-promoting effect of B4GALT7 in HCC.

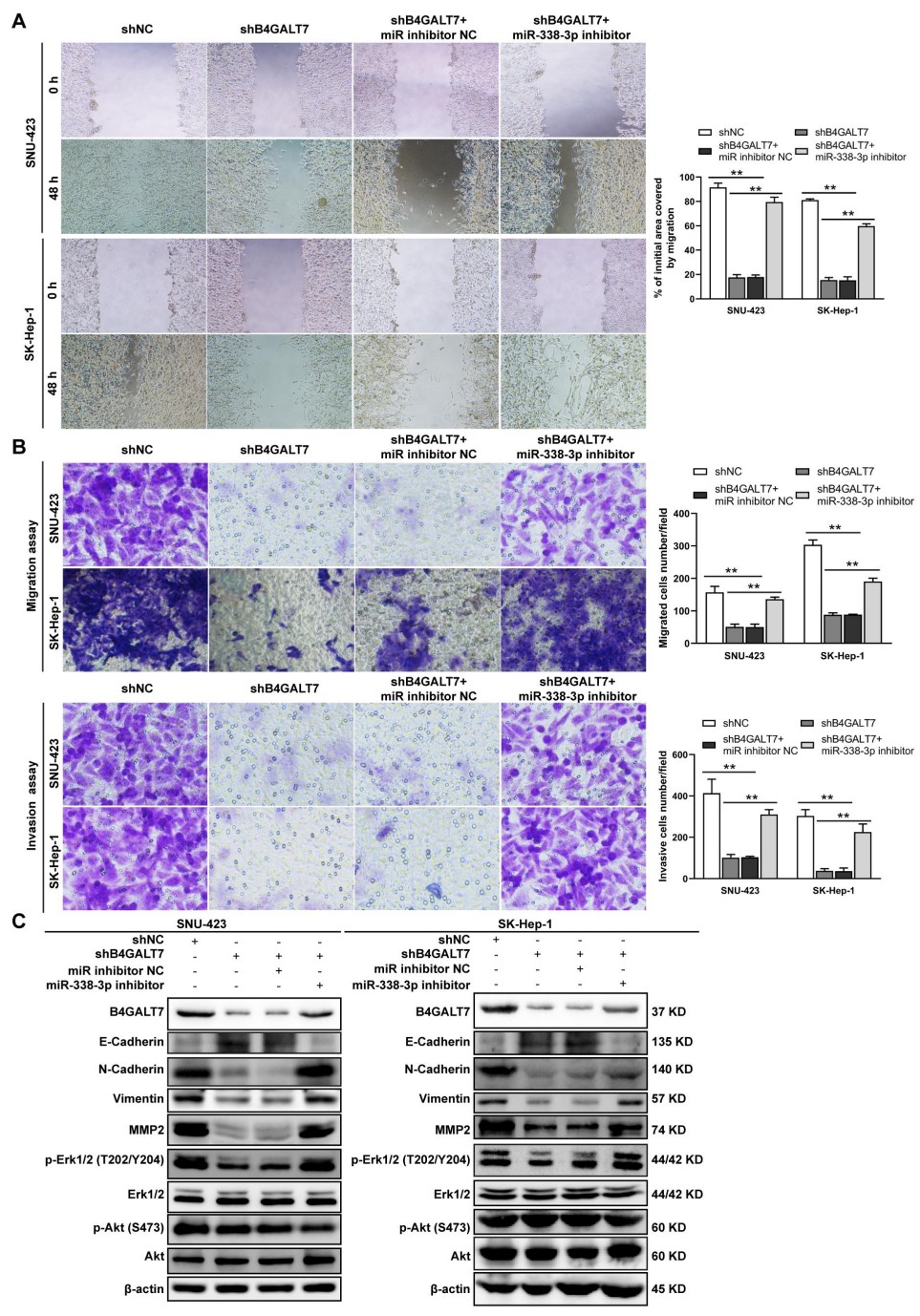

**Figure 7** **The effects of B4GALT7 and miR-338-3p on HCC cell invasion and migration.** (A–B) Wound healing assay, matrigel-free and matrigel-based transwell assays were performed in SNU-423 and SK-Hep-1 cells transfected with shNC, shB4GALT7, and co-transfected with sh-B4GALT7 and the miR-338-3p inhibitor. Migration of the cells to the wound was photographed at 0 h and 48 h. Scale bar: 100 μm. (C) Western blotting assay revealed the expression levels of EMT marker proteins and the phosphorylation status of signaling proteins can be rescued when co-transfected with sh-B4GALT7 and the miR-338-3p inhibitor. *, $P < 0.05$; **, $P < 0.01$.

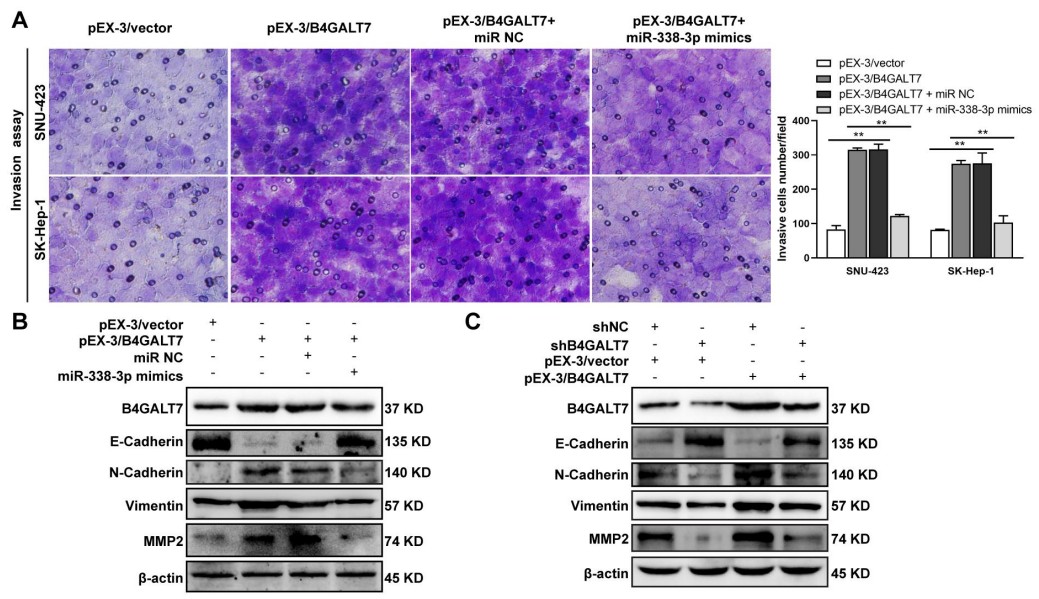

**Figure 8** **The effects of B4GALT7 and miR-338-3p on HCC cell invasion.** (A) Matrigel-based transwell assay was conducted in HCC cells transfected with pEX-3/vector, pEX-3/B4GALT7, and co-transfected with pEX-3/B4GALT7 and miR-338-3p mimics. (B) Western blotting assay revealed the expression levels of EMT marker proteins can be reversed when co-transfected with pEX-3/B4GALT7 and miR-338-3p mimics. (C) Representative western blots of EMT marker protein levels in SNU-423 cells transfected with shB4GALT7 or shNC, and further transfected with pEX-3/B4GALT7 or pEX-3/vector. *, $P < 0.05$; **, $P < 0.01$.

## DISCUSSION

TME consists of immune cells, stromal cells, endothelial cells, cancer-associated fibroblasts, ECM, vasculature and chemokines (*Yang et al., 2020*). ECM is a three-dimensional architectural network involving GAGs, PGs and other glycoproteins (*Karamanos et al., 2021*). Among them, PGs are elevated when the liver is exposed to stressful injuries (*Váncza et al., 2022*). PGs are involved in cell–cell and cell–matrix interactions consisting of one or more GAG chains attached to core proteins (*Li et al., 2022*). Five glycosyltransferases, encoded by genes XYLT1, XYLT2, B4GALT7, B3GALT6, B3GAT3 catalyze the synthesis of the tetrasaccharide linker region between the core protein and the GAG chain (*Li et al., 2022*). Among them, B4GALT7 is localized in chromosome 5q35.3 with 8 exons and 984 nucleotides in length. B4GALTs (beta 1,4-galactosyltransferases) are a family of glycosyltransferases with seven members that are involved in tumorigenesis (*Dai, Wang & Gao, 2022*; *Shirane et al., 2014*; *Wang et al., 2021*), embryonic development (*Kremer et al., 2020*), immune and inflammatory responses (*Chatterjee, Balram & Li, 2021*; *Liu et al., 2018*). However, the function of most B4GALTs has not been investigated individually.

Abnormal protein glycosylation is correlated with cancer malignant phenotypes due to changed protein function and cell–cell communication (*Dusoswa et al., 2020*). B4GALT7 encodes beta-1,4-galactosyltransferase 7, a transmembrane enzyme with 327 amino acids that catalyzes the attachment of galactose to xylose in the synthesis of tetrasaccharide

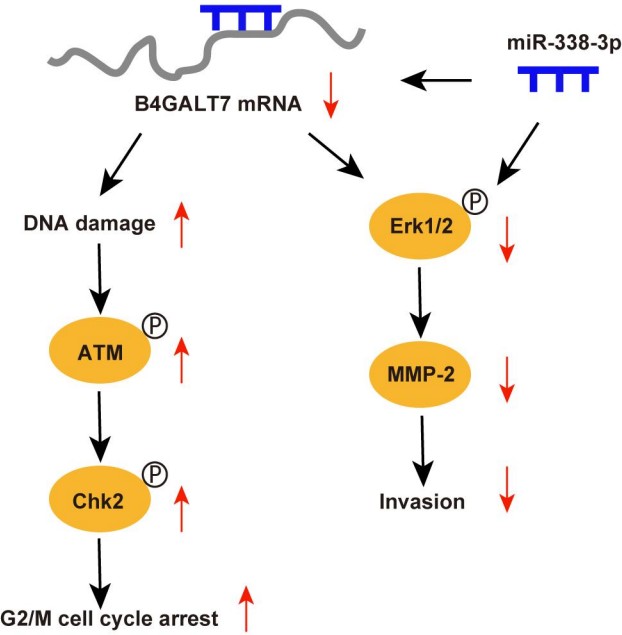

**Figure 9  Schematic representation of B4GALT7 in HCC.**

linkage region of PGs. This enzyme is involved in the O-linked glycosylation-mediated biosynthesis of PGs, a significant component of ECM (*Sandler-Wilson et al., 2019*). In this study, we revealed an oncogenic role for B4GALT7 and a possible modulation by miR-338-3p in HCC development (Fig. 9). The TCGA data indicates that B4GALT7 is expressed at high levels during HCC tumorigenesis, which is correlated with poor prognosis for HCC patients. Consistently, B4GALT7 was upregulated in HCC cell lines and tissues compared with corresponding para-tumor specimens. Using gain-of and loss-of function assays, we revealed that shRNA mediated B4GALT7 suppression reduced HCC cell proliferative, migrative and invasive abilities *in vitro*, but did not affect cell apoptosis obviously. ShRNA mediated B4GALT7 suppression promoted DNA damage and cell cycle arrest at the G2/M phase. We further examined the effect of B4GALT7 suppression on HCC cell growth, migration and invasion *in vivo*. However, both SNU-423 and SK-Hep-1 cells were not suitable for establishing xenograft models. We further discovered that the HCC-promoting effect of B4GALT7 is most likely attributed to B4GALT7-mediated activation of MMP2. Collectively, the above experiments demonstrated that shRNA mediated B4GALT7 suppression reduced cell proliferative, migrative and invasive abilities *in vitro*, and B4GALT7 acted as an oncogene in HCC.

The data indicated that B4GALT7 suppression induced DNA damage and cell cycle arrest at the G2/M phase. DNA damage response involves cell cycle arrest to allow repair (*Smith et al., 2020*). ATM is the crucial regulator in mediating cellular response to DNA double-strand breaks and phosphorylates Chk2 and H2AX to regulate cell cycle arrest and apoptosis (*Smith et al., 2020*; *Cao et al., 2021*). Activated Chk2 further phosphorylates and degrades Cdc25C,

which suppresses phosphorylation of CDKs through phosphorylation of Wee1 (*Smith et al., 2020*). Cdc2 (CDK1) is dephosphorylated and activated by Cdc25 and phosphorylated and inactivated by Wee1 (*Matthews, Bertoli & De Bruin, 2022*; *Elbaek, Petrosius & Sørensen, 2020*). Cdc2 bound to cyclin B1 (*Yuan et al., 2021*) and the Cdc2/cyclin B1 complex is suppressed by Wee1 (*Shin et al., 2019*). Therefore, ATM can regulate G2/M cell cycle arrest *via* Chk2. Consistently, our results indicated that B4GALT7 suppression markedly elevated phosphorylation of ATM at Ser1981, H2AX at Ser139, Chk2 at Thr68, Wee1 at Ser642, Cdc2 at Tyr15, and promoted the level of cyclin B1. The ATM-Chk2-Cdc2/cyclinB1 signaling in HCC cells was elevated after shB4GALT7 transfection, and the cell cycle arrest was rescued after plasmid pEX-3/B4GALT7 transfection. Previous studies suggested that cyclin B1/CDK1 is indispensable for reduction of apoptosis in tumors (*Allan & Clarke, 2007*; *O'Connor et al., 2000*). Consistently, no significant cell apoptosis in HCC cells was observed after shB4GALT7 transfection.

MicroRNAs (miRNAs) are closely correlated with tumorigenesis (*Di Martino et al., 2022*). Previous studies have revealed that miR-338-3p was involved in the progression and EMT of human cancers (*Li et al., 2021a*; *Zhang et al., 2021b*), which was consistent with our data. The dual luciferase reporter assay revealed that miR-338-3p was able to bind the 3′ UTR of B4GALT7 and the reciprocal suppressive effect of B4GALT7 and miR-338-3p was revealed by RT-qPCR and western blotting. Since B4GALT7 and miR-338-3p negatively modulated each other in HCC, we investigated whether B4GALT7 plays a role in the migration and invasion of HCC cells. We discovered that B4GALT7 is associated with the migratory and invasive capabilities of HCC cells. B4GALT7 suppression in indicated HCC cells reduced cell migration and invasion. The suppressive effect of B4GALT7 on HCC cell proliferative and invasive abilities was further reversed by miR-338-3p inhibitor. Consistently, the expression levels of EMT marker proteins and MMP-2, and the phosphorylation levels of signaling proteins were all recovered after co-transfection with shB4GALT7 and the miR-338-3p inhibitor or after transfection with plasmid pEX-3/B4GALT7 in HCC cells. Consequently, these results demonstrated that highly expressed miR-338-3p rescued the tumor-promoting effect of B4GALT7 in HCC.

Reduced B4GALT7 expression downregulated the expressions of MMP-2, mesenchymal markers N-cadherin and vimentin, and upregulated the expression of epithelial marker E-cadherin, which can be further recovered after co-transfection with shB4GALT7 and the miR-338-3p inhibitor. MMPs, particularly MMP2, correlate with EMT during tumorigenesis (*Fan et al., 2021*; *Shi et al., 2021a*; *Shi et al., 2021b*; *Wang et al., 2018*; *Zhao et al., 2022*; *Ye et al., 2021*; *Han et al., 2016*). Previous studies demonstrated that MMP2 was regulated by the PI3K/Akt and the MAP kinase pathways (*Ye et al., 2021*; *Han et al., 2016*). Here, we found a positive correlation between B4GALT7 and MMP2 expression in HCC. We further found that MMP2 expression in indicated HCC cells was significantly reduced upon miR-338-3p mimics transfection, which could be reversed upon miR-338-3p inhibitors transfection.

In conclusion, our *in vitro* study reveals that B4GALT7 is expressed at high levels in HCC and upregulated B4GALT7 expression correlated with HCC invasive abilities. Our data also demonstrate that B4GALT7 suppression might reduce HCC cell invasion by

downregulating MMP2 and the MAP kinase pathway. Moreover, the role of B4GALT7 plays in HCC awaits further validation and exploration.

### Funding

This work was supported by the grants from the National Natural Science Foundation of China (Nos. 30901821, 81172136, and 82072737), the Natural Science Basic Project of Shanxi Province, China (Nos. 20210302124183, 202103021224238, 202103021224240, 201701D121165, 201801D221069, and 201901D111190), the Scientific and Technological Innovation Programs of Higher Education Institutions in Shanxi (No. 2021L339), the Scientific Research Starting Foundation for Doctor of Changzhi Medical College (No. BS202007), the Research Project Supported by Shanxi Scholarship Council of China (Nos. 2020-194, and 2021-165), the Open Fund from Key Laboratory of Cellular Physiology (Shanxi Medical University), the Ministry of Education, China (No. KLMEC/SXMU-202011, CPOF202301), the Shanxi '1331 Project' Key Subjects Construction, China (No. 1331KSC), the Outstanding Youth Foundation of Shanxi Province, China (No. 201901D211547), the Scientific research project of Shanxi Provincial Health Commission, China (No. 2019059), and the "136" College-level open fund, China (No. 2021YZ03). The funders had no role in study design, data collection and analysis, decision to publish, or preparation of the manuscript.

### Grant Disclosures

The following grant information was disclosed by the authors:
National Natural Science Foundation of China: 30901821, 81172136, 82072737.
Natural Science Basic Project of Shanxi Province, China: 20210302124183, 202103021224238, 202103021224240, 201701D121165, 201801D221069, 201901D111190.
Scientific and Technological Innovation Programs of Higher Education Institutions in Shanxi: 2021L339.
Scientific Research Starting Foundation for Doctor of Changzhi Medical College: BS202007.
Research Project Supported by Shanxi Scholarship Council of China: Nos. 2020-194, and 2021-165.
Open Fund from Key Laboratory of Cellular Physiology (Shanxi Medical University).
Ministry of Education, China: KLMEC/SXMU-202011, CPOF202301.
Shanxi '1331 Project' Key Subjects Construction, China: 1331KSC.
Scientific research project of Shanxi Provincial Health Commission, China: 201901D211547.
Scientific research project of Shanxi Provincial Health Commission, China: 2019059.
"136" College-level open fund, China: 2021YZ03.

### Competing Interests

The authors declare there are no competing interests.

## Author Contributions

- Chang Liu conceived and designed the experiments, performed the experiments, analyzed the data, prepared figures and/or tables, authored or reviewed drafts of the article, and approved the final draft.
- Yuqi Jia conceived and designed the experiments, performed the experiments, analyzed the data, prepared figures and/or tables, authored or reviewed drafts of the article, and approved the final draft.
- Xinan Zhao conceived and designed the experiments, performed the experiments, prepared figures and/or tables, and approved the final draft.
- Zifeng Wang conceived and designed the experiments, performed the experiments, prepared figures and/or tables, and approved the final draft.
- Xiaoxia Zhu conceived and designed the experiments, performed the experiments, prepared figures and/or tables, and approved the final draft.
- Chan Zhang conceived and designed the experiments, performed the experiments, prepared figures and/or tables, and approved the final draft.
- Xiaoning Li conceived and designed the experiments, authored or reviewed drafts of the article, and approved the final draft.
- Xuhua Zhao conceived and designed the experiments, authored or reviewed drafts of the article, and approved the final draft.
- Tao Gong conceived and designed the experiments, authored or reviewed drafts of the article, and approved the final draft.
- Hong Zhao conceived and designed the experiments, authored or reviewed drafts of the article, and approved the final draft.
- Dong Zhang conceived and designed the experiments, authored or reviewed drafts of the article, and approved the final draft.
- Yuhu Niu conceived and designed the experiments, authored or reviewed drafts of the article, and approved the final draft.
- Xiushan Dong conceived and designed the experiments, authored or reviewed drafts of the article, and approved the final draft.
- Gaopeng Li conceived and designed the experiments, authored or reviewed drafts of the article, and approved the final draft.
- Feng Li conceived and designed the experiments, authored or reviewed drafts of the article, and approved the final draft.
- Hongwei Zhang conceived and designed the experiments, authored or reviewed drafts of the article, and approved the final draft.
- Li Zhang conceived and designed the experiments, authored or reviewed drafts of the article, and approved the final draft.
- Jun Xu conceived and designed the experiments, authored or reviewed drafts of the article, and approved the final draft.
- Baofeng Yu conceived and designed the experiments, analyzed the data, authored or reviewed drafts of the article, and approved the final draft.

## Human Ethics

The following information was supplied relating to ethical approvals (*i.e.*, approving body and any reference numbers):

The ethics committee of the Affiliated Tumor Hospital of Shanxi Medical University (Ethical Application Ref: KY2023017).

## Data Availability

The raw measurements are available in the Supplementary Files.

## Supplemental Information

Supplemental information for this article can be found online at http://dx.doi.org/10.7717/peerj.16450#supplemental-information.

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
