# Peer review of "Targeting B4GALT7 suppresses the proliferation, migration and invasion of hepatocellular carcinoma through the Cdc2/CyclinB1 and miR-338-3p/MMP2 pathway"

_PeerJ, doi:10.7717/peerj.16450_

## Round 0.1 · original submission · Minor Revisions

Thanks for submitting your work. I suggest you read carefully the comments and the issues pointed out by the reviewers and address them before resubmitting.

Reviewer 1 ·

Basic reporting

1. In Line 55, Could the authors clarify what it means to identify the "mechanism" of a protein?

2. In Line 66, could the authors provide appropriate citations?

Experimental design

NA

Validity of the findings

1. With respect to Fig. 1A, The Methods section highlights retrieval of 10 HCC and para-tumor specimen pairs. Did the authors observe up-regulation of B4GALT7 in 3/10 pairs only? Could the authors show data for all 10 pairs?

Additional comments

This manuscript aims to investigate the role of B4GALT7 in HCC by investigating the RNA-level and protein-level expressions in HCC tumor samples and HCC cell lines. The authors also perform knock-down studies in vitro towards this end and observe reduced proliferation rates in HCC cell lines. Overall the work done here is valid, and the conclusions agree with the observations. I have left my specific comments in the sections above.

Reviewer 2 ·

Basic reporting

The authors presented a study delving into the role of galactosyltransferase in Hepatocellular Carcinoma (HCC). Given the established significance of the extracellular matrix (ECM) in tumorigenesis, the focus on B4GALT7, a gene encoding for a galactosyltransferase, is timely and relevant. The manuscript is well-organized, and the content is presented clearly.

Experimental design

The methodology section is detailed, providing a comprehensive overview of the techniques employed, seeks to elucidate the oncogenic functions of B4GALT7 in HCC.

Validity of the findings

The results section is coherent and supports the hypothesis set out in the introduction. Figures are appropriately used to illustrate key findings, enhancing the manuscript's clarity. However, I noticed a possible discrepancy. In the Introduction, line 63, the manuscript states that “B4GALT7 (beta-1,4-galactosyltransferase, polypeptide 7) encodes galactosyltransferase I”, whereas in common literature, this nomenclature might differ. It would be beneficial to confirm and ensure the accuracy of this classification. Also, there are some suggestions as follows.
1. In the introduction, please elaborate on the pathway that is explored in this manuscript, as certain elements presented in the results and discussion section seem more appropriate for the introductory segment.
2. In the results, try to adjust the borders of the figures to be consistent. Try to manage the size of each figure to make them reader friendly.
3. Akt phosphorylation is pivotal in the PI3K/Akt pathway, serving as an indicator of its activation status. If Akt's phosphorylation remains unaltered, it suggests that this central event in the PI3K/Akt pathway was not modulated. It would be beneficial for the author to present a comprehensive overview (a pathway graph) of the studied pathway in the manuscript, elucidating the interrelation and impact on each component.

---

## Round 0.2 · Minor Revisions

I thank the authors for their work on the manuscript.
I still have some minor comments to the text that I kindly ask the author to take into consideration. Please see the notes in the file attached.

Reviewer 2 ·

Basic reporting

It appears that some of the figures were produced using online tools, resulting in inconsistencies in style compared to the others. Try to enhance this as much as possible.

Experimental design

The experimental design met the standard.

Validity of the findings

The author has addressed the comments and concerns. The manuscript has been refined.

---

## Round 0.3 · Minor Revisions

Thanks for the modifications made to the manuscript. I acknowledge the work of the authors, nevertheless, I still think that the final part of the introduction describes too much in details the work that will be presented in the manuscript. The discussion of the pathways invwstgated with the relative results should be separately presented later in the manuscript. I recommend these modifications beffore acceptance of the work.

---

## Round 0.4 · Minor Revisions

Thanks again for your work. Please find in the attached file a suggestion of the type of comment a reader would expect at the end of the introduction, which is the declaration of intent of the study. By providing all the information about results in the introduction, the reader is not moved to get through the paper.

Please move all the other information (if not already present) to the results section.

---

## Round 0.5 · accepted · Accept

I acknowledge the comments have been addressed.

Since all the previous concerns from the reviewers were already implemented I am pleased to accept this manuscript